Genomic analysis of morphometric traits in bighorn sheep using the Ovine Infinium® HD SNP BeadChip

Miller Joshua M. jmm1@ualberta.ca joshua.miller@yale.edu 1 3
Festa-Bianchet Marco 2
Coltman David W. 1
1 Department of Biological Sciences, University of Alberta , Edmonton , Alberta , Canada
2 Département de biologie, Université de Sherbrooke , Sherbrooke , Québec , Canada
3 Current affiliation:  Department of Ecology and Evolutionary Biology, Yale University , New Haven, CT , USA
Eguiarte Luis
Electronic publication date: 2018 Feb 12
Publication date: 2018
Volume: 6
Electronic Location ID: e4364
Received 2017 Sep 22; Accepted 2018 Jan 23
Copyright: ©2018 Miller et al.
Copyright year: 2018
Copyright holder: Miller et al.
License: This is an open access article distributed under the terms of the Creative Commons Attribution License, which permits unrestricted use, distribution, reproduction and adaptation in any medium and for any purpose provided that it is properly attributed. For attribution, the original author(s), title, publication source (PeerJ) and either DOI or URL of the article must be cited.
License URL: https://creativecommons.org/licenses/by/4.0/

Keywords: GWAS, SNP, Animal model, Ovis canadensis

Funding: National Science and Engineering Research Council (NSERC) Discovery Alberta Conservation Association Grants in Biodiversity Alberta Fish & Wildlife NSERC Discovery Alberta Sport, Recreation, Parks, and Wildlife Foundation Development Initiatives Program NSERC Vanier scholarship Killam Foundation Alberta Innovates Technology Futures Field work at RM has been supported by National Science and Engineering Research Council (NSERC) Discovery Grants, Alberta Conservation Association Grants in Biodiversity to Marco Festa-Bianchet. Alberta Fish & Wildlife provide logistic and financial support. The molecular work was supported by an NSERC Discovery Grant to David Coltman, as well as an Alberta Conservation Association Grant in Biodiversity, and an Alberta Sport, Recreation, Parks, and Wildlife Foundation Development Initiatives Program grant to Joshua Miller. Joshua Miller’s graduate research was supported by an NSERC Vanier scholarship, the Killam Foundation, and Alberta Innovates Technology Futures. The funders had no role in study design, data collection and analysis, decision to publish, or preparation of the manuscript.

==============================
Elucidating the genetic basis of fitness-related traits is a major goal of molecular ecology. Traits subject to sexual selection are particularly interesting, as non-random mate choice should deplete genetic variation and thereby their evolutionary benefits. We examined the genetic basis of three sexually selected morphometric traits in bighorn sheep (Ovis canadensis): horn length, horn base circumference, and body mass. These traits are of specific concern in bighorn sheep as artificial selection through trophy hunting opposes sexual selection. Specifically, horn size determines trophy status and, in most North American jurisdictions, if an individual can be legally harvested. Using between 7,994–9,552 phenotypic measures from the long-term individual-based study at Ram Mountain (Alberta, Canada), we first showed that all three traits are heritable (h2 = 0.15–0.23). We then conducted a genome-wide association study (GWAS) utilizing a set of 3,777 SNPs typed in 76 individuals using the Ovine Infinium® HD SNP BeadChip. We found suggestive association for body mass at a single locus (OAR9_91647990). The absence of strong associations with SNPs suggests that the traits are likely polygenic. These results represent a step forward for characterizing the genetic architecture of fitness related traits in sexually dimorphic ungulates.

Introduction

A goal of molecular ecology is to identify the genetic architecture underlying traits of ecological relevance (Ellegren & Sheldon, 2008; Slate et al., 2009). There is a particular interest in finding regions associated with variation in fitness-related traits, as such traits and the loci underlying them are expected to be subject to strong selection. Under strong directional selection, the genetic variability underlying fitness-related traits should rapidly go to fixation, and yet much phenotypic variation in such traits is observed in the wild (Kruuk, Slate & Wilson, 2008; Chenoweth & McGuigan, 2010). Elucidating the genetic basis of fitness-related traits might help clarify how phenotypic variation is maintained, for example by overdominance or epistatic interactions (e.g., Lappalainen et al., 2011; Johnston et al., 2013).

Sexual selection poses particularly interesting scenarios for fitness-related traits, leading to the so called lek paradox (Borgia, 1979). Under sexual selection, mate choice is either non-random, based on an ornamental trait that may confer benefits to the offspring, or members of one sex compete for access to mates using dimorphic secondary sex characteristics. Over time, under both scenarios evolution should deplete the genetic variation underlying the selected trait, and thereby diminishing offspring benefits. Paradoxically, however, in many systems choice for such traits continues. A classic example of a secondary sexual characteristic often subject to sexual selection is horn size in bovids. Horns are characterized by a keratin sheath around a bony projection from the skull that grows continuously throughout life (Davis, Brakora & Lee, 2011). Across a number of species, horn size in males determines social status and mating access to females (Bro-Jørgensen, 2007). Thus, there is likely selection for variants that confer the ability of individuals to grow large horns, and yet variation remains.

In domestic sheep (Ovis aries) some breeds have horns while others are polled (lacking horns entirely), and from an agronomic production standpoint there is interest in removing horns from certain breeds (Kijas et al., 2012). Soay sheep, a primitive breed now living feral on the islands of St. Kilda, Scotland, have an additional horn “morph”. In females, there are three morphs: normal, scurred (deformed horns composed only of keratin sheaths), and polled. In males there are only two morphs: normal horns and scurs (Johnston et al., 2010; Johnston et al., 2011).

The genetic basis of these differences in horn morphology and development in domestic sheep has been investigated. A single genomic region on chromosome 10 is associated with the presence and absence of horns in domestic breeds (Kijas et al., 2012) including Soay sheep (Johnston et al., 2010; Johnston et al., 2011), and is linked to quantitative differences in horn length of normal-horned male Soay sheep (Johnston et al., 2011). This region contains a single gene, relaxin/insulin-like family peptide receptor 2 (RXFP2). RXFP2 affects osteoporosis and testicular descent in mice and humans (Ferlin et al., 2008; Feng et al., 2009; Yuan et al., 2010); thus, its association with both bone development and secondary sexual characteristics make it an interesting candidate for influencing horn morphology. Furthermore, different genotypes at this locus in male Soay sheep are associated with trade-offs between reproductive success and longevity, which may maintain the different horn morphs through heterozygote overdominance (Johnston et al., 2013). Finally, though not the major QTL underlying horn phenotype, RXFP2 has been implicated in horn development in several association studies in cattle (Bos taurus) (Gautier & Naves, 2011; Allais-Bonnet et al., 2013; Wiedemar & Drögemüller, 2015), indicating that it may have similar function across species.

Bighorn sheep (Ovis canadensis) are iconic North American wild sheep named for their large horns. All individuals have normal horns, though males have much larger horns than females. Previous research has shown that horn size and body mass are important for intrasexual competition among males for reproductive access to females (Coltman et al., 2002; Martin et al., 2013). However, for female bighorn sheep horn length was found to be unrelated to social rank or other life history characteristics, which were more determined by body mass and age (Favre, Martin & Festa-Bianchet, 2008). In addition, horn size determines the trophy status of a male and, in many jurisdictions, whether it can be legally harvested. Such regulations directly influence longevity, as males with fast-growing horns are removed at a younger age (Festa-Bianchet et al., 2008; Festa-Bianchet et al., 2014; Bonenfant et al., 2009; Hengeveld & Festa-Bianchet, 2011; Pigeon et al., 2016).

Previous studies of bighorn sheep have shown that both horn size and body mass are heritable (Coltman, 2005; Coltman et al., 2005; Poissant et al., 2008; Poissant et al., 2012) and quantitative trait locus (QTL) mapping with microsatellite loci highlighted several suggestive regions for different aspects of horn morphology (e.g., volume and base circumference) as well as body mass (Poissant et al., 2012). These regions appear on several chromosomes, but notably included suggestive QTLs for horn volume and base circumference in males on chromosome 10 that spans the region predicted to contain RXFP2. This same region was highlighted by Kardos et al. (2015) who used pooled genome sequencing to search for signatures of selective sweeps in bighorn sheep. Analyses from pooled samples representing 58 bighorn sheep from three populations across Montana (USA) found 21 regions showing signs of selective sweeps. The strongest signal from these regions contained RXFP2, and a comparison to haplotypes in Soay sheep (Johnston et al., 2013) suggested that the selected alleles are associated with large horn size (Kardos et al., 2015). However, there has yet to be an individual-based study of the genomics of horn size, as Kardos et al. (2015) was based on pooled sequencing, and no assessments have been done of other fitness related traits.

In this study, we build on these results using individual phenotypic data from a long-term study of marked sheep followed throughout their lives at Ram Mountain, Alberta Canada. Though the costs of genome sequencing continue to decline, large-scale resequencing projects are still out of reach for many study systems. Therefore, we capitalize on the close evolutionary relationship between domestic and bighorn sheep (Bunch et al., 2006) to efficiently genotype many SNPs in many individual bighorn sheep using a genomic technology developed for domestic sheep. By investigating a suite of morphological characters, we aim to examine the genetic architecture underlying complex quantitative traits in wild sheep, and with respect to horn size, assess if the architecture is similar to its domestic relative.

Methods and Materials

Population history and phenotypic data collection

Bighorn sheep at Ram Mountain (RM) are native, isolated and philopatric. Individual-based monitoring of the population began in 1972 (Jorgenson et al., 1993; Jorgenson et al., 1997). Most individuals are marked with unique tags as lambs, so over 95% are of known age. Individuals are followed throughout their lives. Every spring and summer, sheep are drawn into a corral trap baited with salt where several phenotypic measurements are taken, including body mass and horn measures (Jorgenson et al., 1993). Genetic sampling of the population began in 1988, and was used to build a genetic pedigree (Coltman et al., 2002; Poissant et al., 2008). In some cases, full or half siblings were inferred from unsampled males using the program COLONY (Wang, 2013). By 2013, the pedigree contained 864 maternal and 528 paternal links involving 1,134 sheep.

Phenotypic measures

This study uses data from animals captured under research protocols that were approved by the University of Alberta Animal Use and Care Committee, affiliated with the Canadian Council for Animal Care (Certificate 610901). We considered three morphological characteristics: average horn length, average horn base circumference, and body mass. Specifically, sheep were weighed to the nearest 250 g with a Detecto spring scale, while horn length (measured along the outside curvature) and base circumference were measured to the nearest millimeter with tape. Each trait measurement was standardized to a sex and age specific standard deviation of one (value divided by the SD for that sex in that age class). We only considered individuals aged 1 or greater to avoid maternal effects (Wilson, Kruuk & Coltman, 2005; Poissant et al., 2012), and pooled males aged ≥9 years and females aged ≥14 years to increase sample sizes in those age classes. Fewer than 10% of either sex lives to these ages (Loison et al., 1999).

Quantitative genetic analyses for morphological characteristics

Quantitative genetic variation in our morphological characteristics was estimated using a series of ‘animal models’. Animal models are linear mixed effects models that incorporate pedigree information along with phenotypic measures to partition phenotypic variation (Vp) into that due to additive genetic variation (Va), permanent environmental effects (Vpe), and residual variation (Vr) (Kruuk, 2004; Wilson et al., 2010). For our analyses, fixed effects included sex, age (as a factor), date of measurement (as a continuous, second-order polynomial), as well as all interactions between the three variables. Random effects were individual identity to account for permanent environmental effects associated with having repeated measures of individuals (Vpe), as well as year of birth (Vyb) and year of measurement (Vy) to account for environmental effects. Thus, phenotypic variation was broken into five components Vp = Va + Vpe + Vy + Vyb + Vr.

The three morphological traits were modeled independently using univariate animal models run in ASReml version 3.0 (Gilmour et al., 2009), based on measurements taken between 1972 and 2012. To maximize statistical power, we considered both sexes simultaneously. Combining the sexes into a single model is justified as cross-sex genetic correlations were either large and positive (1.00 for horn length and 0.76 for body mass) or positive but not significant (0.03 for horn base circumference) indicating that we should capture the same genetic variation in both sexes (Poissant et al., 2012). The effect size of each random effect was calculated as the proportion of Vp explained by the random effect, and its significance tested by comparing a model with the term removed to the full model using a likelihood ratio test with one degree of freedom. From these models, we calculated heritability (h2) of each trait as the ratio of Va∕Vp. We also recorded estimates of individual breeding values (Va), calculated using best linear unbiased predictors (BLUPs), for use in selecting individuals for the association analyses (see below).

SNP genotyping

We chose 95 individuals for genotyping based on their breeding value for horn length. Specifically, we attempted to maximize our chances of detecting an association by choosing an approximately equal number of individuals of each sex with the highest and lowest breeding values with respect to horn length (Li et al., 2011; Barnett, Lee & Lin, 2013). The range of high values was 0.18 to 0.65 for males and 0.16 to 0.44 for females, while the range of low values was −0.41 to −0.81 for males and −0.35 to −0.61 for females. The selected individuals were typed on the Ovine Infinium® HD SNP BeadChip an array originally developed for domestic sheep that contains 606,006 loci distributed throughout the domestic sheep genome (Kijas et al., 2014). Initial assessment of genotype quality was performed using Genome Studio version 2011.1 (Genotyping Module 1.9; Illumina, San Diego, CA, USA). We used cluster information based on 288 domestic sheep samples representing a diversity of breeds (provided by the International Sheep Genomics Consortium) and discarded all loci with GenCall scores less than 0.6 and GenTrain scores less than 0.8. GenCall and GenTrain scores are calculated in Genome Studio as quality measures of individual genotypes and locus clustering, respectively. Genotypes were then exported to PLINK version 1.07 (Purcell et al., 2007) for additional filtering. Specifically, we considered only those loci which mapped to the autosomes in domestic sheep, had a minor allele frequency >5%, and were in Hardy-Weinberg Equilibrium (adjusted p >1.28 × 10−5) in our sample set (N = 3,777 remaining). Finally, we used VIPER (Paterson et al., 2012) to check for evidence of pedigree inconsistencies in our dataset. Specifically, this program implements an inheritance-checking algorithm based on a provided pedigree.

Genome-wide association study (GWAS) analyses

Traditional GWAS methods are not designed for repeated measure data (Rönnegård et al., 2016). Alternatives have included fitting individual average values or using breeding values as the phenotypic measure (Johnston et al., 2011; Santure et al., 2013). However, both methods produce undesirable results including inflated association statistics (Hadfield et al., 2010; Ekine et al., 2014). Therefore, we used an alternative method designed for repeated measure data that simultaneously considers phenotypic and SNP data. Specifically, we used the R package RepeatABEL version 1.8-0 (Rönnegård et al., 2016) an extension of the GenABLE package (Aulchenko et al., 2007; Karssen, Van Duijn & Aulchenko, 2016).

RepeatABEL solves the issue of using repeated measures in GWAS by conducting analyses in two steps. First, a base linear-mixed effect model is fit without SNP effects but including a genome-wide relationship matrix (GRM) to account for polygenic effects and individual ID as a random effect to account for repeated measures. In our analyses, the base model had the same structure as that used in the ASREML analyses above. Second, the estimated (co)variance matrix from the first step is used when individual SNPs are tested for association with the phenotype. Specifically, associations are assessed using a linear model and p-values are calculated with a Wald statistic. We fit separate models for horn length, horn base circumference, and body mass.

To correct for multiple testing we used Keff (Moskvina & Schmidt, 2008) to determine significance thresholds genome-wide, and for each chromosome individually assuming an alpha value of 0.05. Association results were then visualized with Manhattan plots created using the ggplot2 package version 1.0.0 (Wickham, 2009). All analyses were conducted in R version 3.2.4 (R Core Team , 2015).

We examined gene annotations in the domestic sheep genome near suggestive loci (see ‘Results’). To determine the genomic window within which to search, we estimated the ‘half-length’ of linkage disequilibrium (LD) for our marker set, i.e., the inter-marker distance at which LD decreased to half its maximal value (Reich et al., 2001). This value is thought to reflect the extent to which an association between genotypes at a given locus and a QTL can be detected. For this analysis we used PLINK version 1.90b2l (Chang et al., 2015) to calculate pairwise values of r2 between syntenic markers on all chromosomes (n = 370,568 pairwise comparisons). These estimates were then compared to inter-marker physical distance based on map positions from the domestic sheep genome, and half-length was calculated using a custom script which calculated LD decay rate as in Appendix 2 of Hill & Weir (1988).

Results

Average horn length, horn base circumference, and body mass all showed positive phenotypic correlations, with the magnitude much stronger in males than females (Table S1). All three morphological traits also exhibited significant additive genetic variation, with values on par with other studies of this population (Table 1). In total, 95 individuals were genotyped on the SNP chip and used to filter loci based on GenTrain and GenCall scores. One individual was subsequently removed from further analyses after significant (>5%) pedigree inconsistencies were found. Of the original 606,006 loci on the chip, 474,277 returned genotypes in bighorn sheep. Subsequent filtering removed 8,528 loci based on their levels of missing data, 469,822 based on our minor allele threshold, and 127 loci based on HWE equilibrium. The final dataset contained 3,777 loci, with at least 60 markers on each autosome (average ± SD = 145.3 ± 88.6; Table S2). Such reductions in the number of polymorphic loci are expected in cross-species application of SNP chips (Miller et al., 2012). Of the 94 originally genotyped individuals, 76 had morphological measures and were used in subsequent analyses.

Table 1 Proportion of phenotypic variance after having accounted for fixed effects in the full datasets.

Variance components of morphometric traits after having accounted for fixed effects in the full datasets; standard errors generated by the statistical software package ASReml version 3.0 (Gilmour et al., 2009) are shown in parentheses unless otherwise noted.

Trait	Inda	Obsb	Mean (s.d.)	Transformed data mean (s.d)	Vp	h2	Vy	Vyb	Vpe	
Horn length (mm)	652	8,011	27.40 (16.98)	6.62 (2.46)	0.85 (0.04)	0.15 (0.05)**	0.07 (0.02)**	0.10 (0.03)**	0.42 (0.05)**	
Horn base circumference (mm)	637	7,994	17.33 (8.33)	12.00 (4.49)	0.84 (0.04)	0.23 (0.05)**	0.08 (0.02)**	0.11 (0.03)**	0.27 (0.04)**	
Body mass (kg)	677	9,552	58.69 (15.85)	7.39 (2.00)	0.58 (0.03)	0.20 (0.04)**	0.16 (0.03)**	0.07 (0.02)**	0.24 (0.04)**	
Notes.

a Numer of individuals.

b Number of phenotypic measurements.

** P < 0.00001.

Manhattan plots for each trait are shown in Fig. 1 with corresponding QQ-plots. In all cases genomic inflation (λ) was ≤1, indicating that there was no underlying population structure or other factors which could lead to false positive associations (Freedman et al., 2004; François et al., 2016). No loci were associated at the genome-wide significance level to any of the morphological traits examined. One locus, OAR9_91647990, showed suggestive association with body mass (Fig. 1, indicated with a green arrow).

Figure 1 Manhattan plots for morphological characteristics.

Horn length (A), horn base circumference (C), and body mass (E). The blue line represents the genome-wide significance threshold; the red line represents the threshold for suggestive association. Positions are relative to the domestic sheep genome assembly (version 3.1; Jiang et al., 2014). The green arrow indicates the suggestive locus for body mass. Next to each Manhattan plot is the corresponding QQ-plot (B, D, and F), with the genomic inflation factor (λ) and standard error indicated in the bottom right of each plot. The black line shows a 1:1 correspondence while the red line is a regression through the observed data.

As expected, there was a general decrease in LD with increasing inter-marker distance, and half-length was estimated to be 412,834 bp (Fig. 2). Based on this half-length estimate we extracted gene names from the Ovis aries gene set (Oar v3.1, genebuild Mar 2015) within a 413,000 bp window on either side of the candidate marker using BioMart (Kinsella et al., 2011) and Ensembl version 89 (Flicek et al., 2014). This returned two genes: U6 spliceosomal RNA, and ENSOARG00000026555, a long intergenic non-coding RNA. No gene ontology (GO) terms were available for either of these genes, and we do not see an immediate connection with body mass.

Figure 2 Scatterplot of LD estimates versus inter-markers distance.

A non-linear least squares regression line is shown, with the round point indicating the half-length estimate.

Discussion

We examined the genetic bases of three fitness-related characteristics in bighorn sheep. To do so, we utilized a new genomic technology originally designed for domestic sheep to rapidly genotype markers in a wild species, then combined these data with phenotypic measures from a long-term individual-based study. We found one locus with suggestive associations to body mass (Fig. 1). Previous QTL mapping with microsatellite loci for these same traits in the RM population highlighted several candidate regions (Poissant et al., 2012); however, our suggestive locus is not near any of the QTLs described in Poissant et al. (2012). In addition, we found no overlap in location between the locus found here and morphological traits in the domestic sheep QTL database (Hu, Fritz & Reecy, 2007; Hu et al., 2013). While it is possible that the sample sizes used in the Poissant et al. (2012) led to an overestimation of effect sizes due to the Beavis Effect (Slate, 2013) we note that the methods underlying QTL mapping and GWAS analyses are different (Slate et al., 2010). Specifically, QTL mapping relies on informative meioses within a pedigree of related individuals, while GWAS uses linkage disequilibrium between loci. In addition, the sample sizes differed, with fewer individuals included in the work presented here. These differences could influence the associations detected.

It is somewhat surprising that we did not see even a suggestive association between horn morphology and the region surrounding RXFP2 on chromosome 10 given the very strong links seen in both domestic sheep and cattle (Gautier & Naves, 2011; Johnston et al., 2011; Johnston et al., 2013; Kijas et al., 2012; Wiedemar et al., 2014) as well as the suggestive QTL for horn volume in bighorn sheep in this same region (Poissant et al., 2012). However, based on the estimate of half-length (412,834 bp) it appears as if we did not have sufficient marker coverage to adequately test for associations in the horns region. Within our set of loci the closest marker to RXFP2 was 698,861 bp away.

It is interesting that the extent of LD reported here (∼400,000 bp) is an order of magnitude less than found in a previous assessment of LD in bighorn sheep from RM (∼4,000,000 by Miller et al., 2011) using an order of magnitude fewer markers (308 vs. 3,777 loci). Analogous decreases in LD with the addition of markers have been seen in other species including cattle (McKay et al., 2007; Porto-Neto, Kijas & Reverter, 2014), domestic sheep (García-Gámez et al., 2012; Kijas et al., 2014), and flycatchers (Ficedula albicollis; Backström et al., 2006; Kawakami et al., 2014).

In light of our failure to detect genome-wide significant associations, we more formally quantified the expected power of a marker to detect a hypothetical causal QTL given the average minor allele frequency and genome wide critical p-value for the loci in this study. To do so we used an R script developed by Minikel (2012) which implements the QTL association feature of the Genetic Power Calculator (Sham et al., 2000; Purcell, Cherny & Sham, 2003). Specifically, this script estimates the expected power to detect an association given an estimate of the QTL effect size, the number of samples genotyped, and the average level of linkage disequilibrium among markers. For our analyses we varied effect sizes from 0–1.0, sample sizes between 50–500 individuals, and three levels of linkage disequilibrium (0.75, 0.50, and 0.25). This exploration showed that even at extreme effect sizes for the QTL and levels of LD well above what was seen at the half-length estimate (∼0.23; Fig. 2), the number of samples used in our GWAS analyses was likely not enough to have the power to detect all associations (Fig. 3). Note that these simulations assume that unrelated individuals were used in the GWAS, so the presence of related individuals in our test set will boost power slightly. In general, the simulations indicate that our marker coverage likely increased the chance of Type II errors (missing true associations). Similar results were found with simulations and whole genome sequences of collared flycatchers (Kardos et al., 2016). However, we do not believe this diminishes the association observed, as it has no effect on Type I errors (detecting false associations).

Figure 3 Heat maps of expected percent power of a GWAS as a function of sample size and effect size for linkage disequilibrium (LD) estimates of 0.75 (A), 0.50 (B), and 0.25 (C). Light colors indicate higher power to detect associations (D). Dotted red lines correspond to the number of samples used in this study (N = 76).

The power of our association analyses was likely also weakened by the cross-species application of a SNP chip originally derived for domestic sheep. While the two species are closely related (Bunch et al., 2006), and have a highly syntenic karyotype (Poissant et al., 2010), loci were selected for inclusion on the chip based on variability in domestic sheep, leading to ascertainment bias when applied to bighorn sheep (Lachance & Tishkoff, 2013). This bias would also increase the chance of Type II errors, as we are unable to assess bighorn sheep specific variants.

Recent research has suggested that a selective sweep occurred around the RXFP2 region in bighorn sheep (Kardos et al., 2015). In this scenario, multiple generations of sexual selection for large horns led to the fixation of genetic variation in the RXFP2 regions. If true, that fixation would preclude detection of associations in the current study. The region described by Kardos et al. (2015) spans ∼350,000 bp and while the Ovine Infinium® HD SNP BeadChip contains 57 SNPs in this region, none of these loci were polymorphic in our sample of sheep from RM.

Finally, the lack of strong associations could be due to the fact that in this species these complex phenotypes are not single-locus traits. Instead, there may be many loci of small effect that jointly contribute to the phenotype, similar to the “missing heritability” phenomenon seen in many quantitative traits (Manolio et al., 2009; Yang et al., 2010). Other studies of the genetic architecture of complex phenotypes in wild populations have also found that they tend not to be controlled by single loci of large effect, but rather are polygenic (Husby et al., 2015; Bérénos et al., 2015; Kardos et al., 2016; Silva et al., 2017). New methods, such as chromosome partitioning, can now investigate this possibility (Yang et al., 2011; Robinson et al., 2013; Santure et al., 2013). Unfortunately, we cannot utilize chromosome partitioning at this time due to the small number of individuals typed on the 700k SNP chip. Attempts to use this method with our data produced unstable estimates of per-chromosome heritability (results not shown). More broadly, if these traits are truly polygenic it helps to explain how their variation is maintained despite strong directional selection (Rowe & Houle, 1996).

Conclusion

The lack of associations found here highlights the challenges of identifying genes underlying traits in non-model systems. While cross-species application of this SNP chip provided a rapid and affordable way to genotype many loci across a large number of individuals, as high-throughput sequencing costs continue to decline we expect this method to be superseded by those that allow for simultaneous marker discovery and genotyping in the species of interest (e.g., Andrews et al., 2016). Future studies could build on our findings by using high-throughput sequencing to increase the number of loci, individuals, and populations used. Improved genomic resources for bighorn sheep (Coltman, Hogg & Miller, 2013; Kardos et al., 2015; Miller et al., 2015) including whole genome sequence will enable fine mapping of associations, as well as detection of novel associations. Consideration of additional populations will allow for assessing the consistency of associations observed. In addition, haplotype-based analyses (Browning & Browning, 2011) or chromosome partitioning methods (Yang et al., 2011; Robinson et al., 2013; Santure et al., 2013) can detect novel associations and highlight if the traits fit a polygenic framework.

Supplemental Information

Table S1 Correlation coefficients among the morphological traits examined in this study

Correlations among females (aged 4 or greater) are below the diagonal, while those for males (aged 4) are above the diagonal. Note that we present the sexes separately given the sexual dimorphism in this species, and these ages as they allow consideration of adult phenotypes (removing autocorrelation with age), while avoiding the effects of selective harvest due to trophy hunting in males as measurements are collected before the hunting season and no male younger than 4 years has ever been legally harvested.

Click here for additional data file.

Table S2 Number distribution of markers used in the GWAS analysis

Genotyping was done using the Ovine Infinium® HD SNP BeadChip an array originally developed for domestic sheep containing 606,006 loci. Marker positions were taken from the domestic sheep genome assembly (version 3.1).

Click here for additional data file.

Supplemental Information 1 R script for repeated measures GWAS analysis using RepeatABEL package

Click here for additional data file.

Supplemental Information 2 Genotypes used in GWAS analysis

tped formatted genotypes for use in GWAS analyses.

Click here for additional data file.

Supplemental Information 3 tfam file for use in GWAS analysis

Click here for additional data file.

Supplemental Information 4 Base circumference measurements for use in GWAS analysis

Click here for additional data file.

Supplemental Information 5 Repeated measures base circumference measurements for use in GWAS analysis

Click here for additional data file.

Supplemental Information 6 Repeated measures horn length measurements for use in GWAS analysis

Click here for additional data file.

Supplemental Information 7 Repeated measures body mass measurements for use in GWAS analysis

Click here for additional data file.

We would like to first and foremost acknowledge the numerous Alberta Fish and Wildlife biologists, graduate students and field assistants who have collected the long-term phenotypic data that went into this work, in particular Jon Jorgenson and Chiara Feder. We acknowledge the contribution of James Kijas and Russell McCulloch at CSIRO for performing SNP array genotyping using the ovine HD SNP chip. Corey Davis and René Malenfant provided thoughtful discussion about analyses and comments on the manuscript.

Additional Information and Declarations

Competing Interests

Author Contributions

Animal Ethics

Data Availability

David W Coltman is an Academic Editor for PeerJ.

Joshua M. Miller conceived and designed the experiments, performed the experiments, analyzed the data, wrote the paper, prepared figures and/or tables, reviewed drafts of the paper.

Marco Festa-Bianchet contributed reagents/materials/analysis tools, wrote the paper, reviewed drafts of the paper.

David W. Coltman conceived and designed the experiments, contributed reagents/materials/analysis tools, wrote the paper, reviewed drafts of the paper.

The following information was supplied relating to ethical approvals (i.e., approving body and any reference numbers):

This study uses data from animals captured under research protocols that were approved by the University of Alberta Animal Use and Care Committee, affiliated with the Canadian Council for Animal Care.

The following information was supplied regarding data availability:

SNP genotypes and morphological measurements are available from the Dryad Digital Repository: https://doi.org/10.5061/dryad.c0p090f.

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
