# Peer review of "Genomic analysis of morphometric traits in bighorn sheep using the Ovine Infinium® HD SNP BeadChip"

_PeerJ, doi:10.7717/peerj.4364_

## Round 0.1 · original submission · Major Revisions

· Academic Editor

Major Revisions

The reviewers agree that this is an interesting data set, but they all have made relevant suggestions and comments. Please, review and answer each one of their comments and points.

Reviewer 1 asks for drastic changes in the aims and analyses of the paper and to remove figures 2 and 3 from the main body of the paper. I do not agree with them in these changes, but the reviewer has other relevant comments for the paper that should be carefully attended.

Reviewer 2 suggest additional selection test that can be conducted, and mentions a relevant references; also indicates that the title of the paper should be improved to describe better the study (to something like “ Exploring the genomic basis of morphometric traits and the lack of association between phenotypic and genetic variation in Bighorn sheep”).

Reviewer 3 points out several important issues that need to be clarified, in particular about the cross-species application of SNP chips, ascertainment bias and coverage that may have important effects in summary statistics, in analysis of genetic structure and in selection tests, and reducing the power of selection tests by increasing type I and type II error.

Also, it is important, as this reviewer indicates, that the discussion should include perspectives on how to perform a better analysis, either by focusing in amplifying the region of the RXFP2 gene (previously identified candidate for horn morphology), or by increasing the genome coverage, by using a massive sequencing approach, for instance RAD-seq.

I have some additional comments:

Abstract:
Include actual data: the heritability of the 3 traits, the name of the population analyzed, the number of year of data, the total number of individuals analyzed, the name of the used chip, the number of actually used SNPs and individuals (377 SNPs, 72 individuals), that only one SNPs associated to body weight, which SNP was, the name of the linked genes, the half length distance for the LD, etc.

Line 49: Explain the lek paradox.

Lines 120-122: In many animal species, most traits are correlated with total size (i.e., larger animals have large and wider horns). Are these 3 variables correlated? Give the values for you data set.

Lines 152 to 153: Explain the relevance and meaning of the of the breeding values and of the BLUPs.

Lines 164 and 165: Briefly, explain what GenTrain and GenCall scores do or mean.

Lines 170 to 177: Explain how RepeatABEL solves the problem of the repeated measures.

Lines 189: Remind the readers of the traits. You could give here the correlations values I asked above.

Lines 192 to 193: Explain what VIPER does, and move this to Methods.

Line 291: Explain what is “genomic inflation” and the meaning of lambda and of the mentioned value.

Lines 218 to 222: All measured with the same half-length method? Explain. Also all this is Discussion, and should be moved there.

Lines 221: Put “Ficedula albicollis” in italics.

Figure 1: I do not see the dashed and solid lines mentioned, but there are one blue and one red lines, that are not explained.
I suggest that you mark the significant SNP with an arrow.
In the legend, give the reference of the 3.1 genome assembly.
Also in the legend, explain what are the squared chi´s off, and what are their black and red lines, the meaning of the included lambdas and of the +- values.

In this figure, there is one grey SNP in body mass that is touching the blue line. It would be interesting also to mention and discuss to what genes this SNP is associated with, considering that these analyses only suggest possible candidate genes.

Figure 3 needs more explanation, as the reviewers indicated. It would be important to mark in the figure the number of samples you had and the average LD you found in the data, to stress the need of more markers and individuals to get significant results.

Table 1: In the heading explain and give a reference of what ASRml is.
In this table you could include the correlations among the 3 variables.
Indicate the units of the Mean (cm, kg).
Explain what are the values and the units of “Year”, “Year of Birth” and “Perm. env”, and the meaning of the last one.

Reviewer 1 ·

Basic reporting

In this study, Miller et al. analyze the genetic basis of three sexually selected traits in a population of bighorn sheep that has been monitored for a long time. The authors use an Ovine Infinium BeadChip and morphologic data to perform genome wide association studies (GWAS) and identify SNPs that could be associated to these traits. The authors find a single SNP showing a low association with body length, and argue that the lack of association suggest that the traits should be controlled by many loci of low effect, instead of single loci of strong effect.
While the objective of the manuscripts is interesting, and the long term monitoring is very valuable and unique, I think that the authors could analyze with more depth the long-term dataset that they have. The main focus of the study is centered on the GWAS aspect of the results, even though they do not find any significant results. The authors argue that the lack of association is probably an effect of the phenotypic traits being regulated by many loci of small effect instead of single loci of strong effect. However, it is also possible that the lack of associations can be related to a low coverage of SNPs. I know that it can be difficult to have large amount of genomic data, and I do not believe this is a main problem in the manuscript. Given the lack of significant results, I think that this paper would be more interesting if they analyzed with more depth the genomics of populations of these individuals, including all the demographic information they have (genetic diversity, paternity analyses, etc…). In addition I would develop more the phenotypic analyses, perhaps also considering the demographic data they have. The authors have a very interesting set of data that could be incorporated into the analyses that would greatly improve the paper, instead of focusing only in the genomic part. In this regard, I have the feeling that in order to have more information, the authors perform LD analyzes and simulations to test for the power of their analyses. While these results are interesting, all this information is supporting the main results and not part of the objectives. I think analyzing with more detail the genomics of this population, and moving figures 2 and 3 to supporting information would make more sense.
I addition I have a few minor comments:
Lines 41-42: “For example by overdominance…..2013)”: The authors do not test or comment any of these examples in the discussion. I think this sentence should be removed.
Line 49: “Lek paradox”: The authors should explain what is the lek paradox, and how would the lek paradox apply for horns.
Lines 54-61: “The genetic basis … (Ovis aries)”: The authors mention that the genetic basis have been studied in this system. Then it is mentioned that breeders have been interested in removing the horns, and finally the authors explain other examples of horns. However the genetic information is explained later. I think it would be more logic to move this sentence to the part where the authors explain the studies that have analyzed the genetic basis in line 62.
Line 71: though should it be through?

Lines 97-99: “Found 21 regions …. (Kardos et al. 2015)”: Genomic studies on horns have already been done. I think it is important to here explain what is missing and why their study could be important. Mention that other traits remain to be analyzed?

Lines 106-107: “we contribute to knowledge on the genetic basis of complex quantitative traits”: I’m not sure this is true, since they only find one SNP with low association. As I mention before, I think it would be more interesting to also analyze the genomic diversity of the population, and focus more on the phenotypic aspect of the manuscript. If not, at least explain how they contribute to this knowledge because it is not clear for me.

METHODS:
Line: 113: “Most individuals are marked … known age”: I think this is very valuable and could be analyzed with more detail. Maybe test if consanguinity changes between generations, or something of this kind. Also comparing morphologic data between generations could be interesting. Exploiting this information could be valuable for the manuscript and build upon the genomic analyses.

Lines 151- 152: “ We also estimated ….BLUPS)”: what is individual breeding values? The authors use this to select the individuals that are genotyped, but for somebody who is not familiar with the concept it would be important to explain it.

Line 167-168: The authors explain which SNPs are discarded, but it would be convenient to also mention the final number of SNPs, so the readers know the amount since the methods and not wait until the results.

Lines 178-183: The authors mention two steps, but they never explain whether the two steps are compared, connected or what is the point of having two steps. Is the first step used to control for the second step?


RESULTS:

Line 191: “used to quality filter loci” odd sentence, the authors should change it.

The authors remove SNPs based on missing data and MAF: What happens if they include some of the discarded SNPs. Maybe it is worth testing the same analyses using other SNPs in the genome. These analyses could go in supporting information, but might increase the power.

Line 201: ”genomic inflation”. Can the authors explain this?

Line 204: “examined” should be examine the

Figure 2 and lines: 204-215: This LD analyses are interesting, but I do not think that the result presented in figure 2 is a main conclusion or result. It is part of the methods and should go in supporting information. I would recommend rather putting a figure showing summary statistics or presenting the phenotypic data in another way.

Lines 226-228: Two genes….not see an immediate connection with body mass”: I think this sentence should go in the discussion and that they should discuss this further. Explain why there is not a connection with body mass, and also explain why they found this association then.
Discussion:
Line 230: “several”, It is only three characteristics, this is not several.
Lines 233-234: “We found 1…. Poissant et al. (2012)”: It is striking that they authors do not find any SNPs within the regions found by those described by Poissant. SNPs should have a larger coverage of the autosomes. I think this needs to be addressed with more detail. Is it possible to map the candidate regions of Poissant and see if any SNP is found nearby? If so, all the candidate regions of Poissant could be false positives. Please develop further this result.

Lines 243-245: “However, based on … horns region”: The authors mention that they did not find a SNP in the region of RXFP. Maybe this is because they lost many SNPs that did not pass their filters. Could the authors verify if any of the SNPs from the chip appear in this region and analyze what happens if they include them in the analyses (Even if they are not in HW, or have a lot of missing data)? As they mention later, it is also probable that some SNPs are fixed in this population. Maybe it would be interesting if the analyze this. The sentence in lines 262-268 could be moved to this part.

Lines 247- 261: As with the LD analyses, the power analysis is not the main focus of the paper. This is interesting but not related to their objectives. These results could be moved to supporting information. Also, it would be important to explain better how they did the analyses, and explain better the parameters changed, because it is not very clear.

Experimental design

As I explain above, I think that it would be valuable if the authors develop more the genomics of populations and the phenotypic analyses. Also it would be interesting to include analyses that consider explicitly the long term study.

Validity of the findings

As I mention above, I think that the paper is too focused on the GWAS analyses, and that I would be more valuable to include other genomic analyses.

·

Basic reporting

No comment

Experimental design

No comment

Validity of the findings

No comment

Comments for the author

In this paper the authors analyzed the genetic basis of three phenotypic features of bighorn sheep. They collected measures of horns and body mass from several individuals. Then, 95 individuals (76 used) were genotyped using a commercial SNP chip. The results showed no association for loci and phenotype (except one). The authors suggested that the results could be due to low sample/loci number or selective sweeps. The conclusions were referred to the need to increase the sample size as well as loci number.

Despite the negative results, I found this manuscript relevant and also well written. Although I have not expertise on QTL analyzes, I found the Methods well explained and in general understandable. However, the authors focused on the negative results, thus leaving aside interesting results.

-They showed the importance of the number of markers (lines 216-218)
-They actually found a locus having significant association, although without GO (lines 203-228)
-They suggested that there could be a selective sweep event that led to the lack of variation on the previous reported loci (line 262)

Considering the above mentioned, I suggest the following:

-There should be carried out tests of selection. This could give insights about the lack of variation found. If a signal is founded, it could be associated with sexual selection and/or hunting pressures, both of them also influenced by population demographics affected y anthropogenic factors.

Please check this paper for a compilation of methods of cattle breeds (Table 1)

Gutiérrez-Gil B, Arranz JJ and Wiener P (2015) An interpretive review of selective sweep studies in Bos taurus cattle populations: identification of unique and shared selection signals across breeds. Front. Genet. 6:167. doi: 10.3389/fgene.2015.00167

-The paper should be complemented with a discussion about the biological factors that could lead to the observed results (as sexual and artificial selection, genetic bottlenecks, etc.), and not rely only in the assumption of the lack of samples/loci.

-Also there should be included reasons why the locus OAR9_91647990 could not be annotated. It was because a lack of well annotated genome? Or the body mass could be affected by regulatory regions that are unknown? The locus in question could have strong linkage with other locus? Are there reports of this?

-Finally, the title is ambiguous. I suggest that it should include something like "..the lack of association of phenotypic and genetic variation...".

Specific comments:

Some parts of the Results section should be moved to Methods.
Lines [192-193] (VIPER Software)
Lines [200] (Manhattan plots, add an explanation of how they were generated)
Lines [204-213] (PLINK Software)


Fig 1
Indicate the meaning of red and blue lines
Fig 2
Add an explanation for the gray scale

·

Basic reporting

The article is interesting as the authors examined three sexually selected traits in bighorn sheeps using a SNP chip designed for domestic sheep and try to idendify candidate genes associated to these traits. I commend the authors for their extensive morphological data set, compiled over many years of field work.The paper is written in clear and professional English, nevertheless there are a few typos that need to be checked and corrected (see detailes below).
The literature reference provide sufficient field background. The introduction is clear, but there are a few concepts that should be explained for clarity such as lek paradox (line 49) and polled morphotype (line 69).
The article has a professional article structure, figures and tables. Raw data is shared. Nevertheless, supplemental materials needs more descriptive metadata identifiers to be useful to future readers.
Overal, the article is interesting. My main concern is that when using cross-species application of SNP chips, ascertainment bias and coverage have important effects in summary statistics, analysis of genetic structure and selection tests that reduce the power of selection tests and increase type I and type II error (see Lachance and Tishkoff, 2013 Bioessays 35:780-786). Therefore, the analysis and discussion of results should address if ascertainment bias was measured and corrected.
In addition, there are a few aspects on the biology of the species and methods used that should be included in the discussion (see below).

Experimental design

This article shows original primary research with a well defined question, which is relevant and meaningful. In addition, it is stated how research fills an identified knowledge gap.
The investigation is interesting, but as mentioned above, when using cross-species application of SNP chips, ascertainment bias and coverage reduce the power of selection tests and increase type I and type II error. Therefore, these issues should be addressed properly before acceptance.
In general, methods are well described. Nevertheless, there are a few aspects that should be described with more detail:
In the methods section (line 166) the authors mention that "Genotypes were then exported to PLINK version 1.07 (Purcell et al., 2007) for additional filtering". Please explain more thoroughly what type of filtering was used and the parameters.
There are a couple of paragraphs (lines 204 - 213 and lines 223 - 225) in the results section that describe methods and should be moved to the methods section.

Validity of the findings

The discussion should include perspectives on how to perform a better analysis, either by focusing in amplifying the region of the RXFP2 gene (previously identified candidate for horn morphology) or increase genome coverage by using other massive sequencing approach such as RAD-seq.
Also, I think it is important to consider if including available data from domestic sheep will help assess asceirtanment data or even analyse if alleles fixed in the bighorn sheep population correlation with adaptive traits.
In the Method section the authors mentioned that the studied bighorn sheep population is isolated and philopatric. These characteristics of the population would result in high inbreeding coefficient and low genetic variation, therefore explaining fixed alleles in the population. I consider that it is important to include such aspects of the population and its effects on the analysis in the discussion.
The authors discuss that the traits they analyzed might be of polygenic inheritance. Nevertheless, this conclusion is speculative and it is importat to propose the kind of analysis needed to test this hypothesis.
Another point that should be addressed in the discussion is the effect of different selection regimes on the results, when there is strong sexual selection towards horn size in males, but not so in females.
Finally, I would recommend that the authors discuss briefly on the utility of SNP chips designed for domestic species in the study of wildlife.

Comments for the author

Please check the following lines:
Line 187: "in in R version 3.2.4". Please remove one "in".
Line 204: "We were interested to examined gene annotations". Should it be "We were interested to examine...".
Line 215: "It interesting to note". Please change for "It is interesting...".
Line 249: "R script developed by (Minikel, 2012)". Please check for the position of parenthesis.

---

## Round 0.2 · Minor Revisions

· Academic Editor

Minor Revisions

I agree with the reviewers that the paper was improved and can be accepted, pending very minor revisions and thank the authors for their efforts.

Please check and correct the suggestions of reviewers 1 and 3, and I have an additional correction:

Lines 128 to 130: aged 1; aged ≥9; aged 14: I think it makes sense it is years, bur please include:
“aged 1 years”; “aged ≥9 years”; “aged 14 years”.

In line 225, I suggest to include: (Figure 1, indicated with a green arrow).

Reviewer 1 ·

Basic reporting

no comments

Experimental design

no comments

Validity of the findings

no comments

Comments for the author

Dear Dr. Eguiarte,
I have now read the new version of the manuscript entitled “Genomic analyses of morphometric traits in bighorn sheep using the Ovine Infinium HD SNP BeadChip” by Miller et al.
I think the authors have addressed well the comments made my editor and the rest of the reviewers. I have three suggestions or concerns that should be attended.
Lines 122-123: Is there a justification of why they measured these three traits and not others. A simple justification about their importance or why they were chosen for this paper would be nice.
Line 218-219: The authors mention that they keep 76 out of the 94 individuals that have measures. Maybe there is something I did not understand well, but they mention in the materials and methods that they chose individuals based on their individual breeding values. Having individual breeding does not mean that you have measures? Maybe there needs to be a better explanation.
Line 229: “candidate markers”: isn’t it only one? In line 224 the authors only mention one candidate locus.

·

Basic reporting

no comment

Experimental design

no comment

Validity of the findings

no comment

Comments for the author

I found this new version pleasant to read. Also, I consider that all suggestions/comments were answered properly.

Therefore, I suggest that this paper should be accepted for publication.

·

Basic reporting

As I mentioned in my previous review, the article is interesting and the paper is written in clear and professional English. I think the authors made a good work addressing my previous comments. I only have minor corrections that should be addressed upon acceptance.

Experimental design

Ok

Validity of the findings

Ok

Comments for the author

I think the authors made a good work addressing my previous comments. I only have minor corrections that should be addressed upon acceptance.


Lines 219 – 222. Report the correlation values published in Poissant et al. 2012.
Line 229 and 231. Indicate the threshold used as breeding value. Define what refers to highest and lowest breeding values.
Line 240. Individual within loci does not makes sense to me. Please revise this.
Line 302. 3,777 instead of 3777.
Line 373. Parenthesis in italics.

---

## Round 0.3 · accepted · Accept

· Academic Editor

Accept

I want to thank the authors for their efforts to improve the paper.

I think this is an interesting contribution to understanding the relationships among SNPs and phenotype and that it is ready for publication.